# Changes in presentations with features potentially indicating cancer in primary care during the COVID-19 pandemic: a retrospective cohort study

Lauren J Scott [ID],[1,2] Mairead Murphy [ID],[2] Sarah Price,[3] Rhys Lewis,[4] Rachel Denholm,[2] Jeremy Horwood,[1,2] Tom Palmer [ID],[2] Chris Salisbury [ID] [2]

[1]National Institute for Health Research Applied Research Collaboration West, University Hospitals Bristol and Weston NHS Foundation Trust, Bristol, UK
[2]Population Health Sciences, University of Bristol Medical School, Bristol, UK
[3]College of Medicine and Health, University of Exeter Medical School, Exeter, UK
[4]One Care, Bristol, UK

**Correspondence to**
Dr Mairead Murphy;
mairead.murphy@bristol.ac.uk

## ABSTRACT

**Objectives** To investigate how the COVID-19 pandemic affected the number of people aged 50+ years presenting to primary care with features that could potentially indicate cancer, and to explore how reporting differed by patient characteristics and in face-to-face vs remote consultations.

**Design, setting and participants** A retrospective cohort study of general practitioner (GP), nurse and paramedic primary care consultations in 21 practices in South-West England covering 123 947 patients. The models compared potential cancer indicators reported in April–July 2019 with April–July 2020.

**Main outcome measures** Potential indicators of cancer were identified using code lists for symptoms, signs, test results and diagnoses listed in the National Institute for Health and Care Excellence suspected cancer referral guidance (NG12).

**Results** During April–July 2019, 17% of registered patients aged 50+ years reported a potential cancer indicator in a consultation with a GP or nurse. During April–July 2020, this reduced to 11% (incidence rate ratio (IRR) 0.64, 95% CI 0.62 to 0.67, p<0.001). Reductions in potential cancer indicators were stable across age group, sex, ethnicity, index of multiple deprivation quintile and shielding status, but less marked in patients with mental health conditions than without (IRR 0.75, 95% CI 0.72 to 0.79, interaction p<0.001). Proportions of GP consultations with potential indicators of cancer reduced between 2019 and 2020 for face-to-face consultations (IRR 0.84, 95% CI 0.76 to 0.92, p<0.001) and increased for remote consultations (IRR 1.17, 95% CI 1.07 to 1.29, p=0.001), although it remained lower in remote consulting than face-to-face in April–July 2020. This difference was greater for nurse/paramedic consultations (face-to-face: IRR 0.61, 95% CI 0.44 to 0.83, p=0.002; remote: IRR 1.60, 95% CI 1.10 to 2.333, p=0.014).

**Conclusion** The number of patients consulting with presentations that could potentially indicate cancer reduced during the first wave of the COVID-19 pandemic. Patients should be encouraged to continue contacting primary care for persistent signs and symptoms, and GPs and nurses should be encouraged to probe patients for further information during remote consulting, in the absence of non-verbal cues.

### Strengths and limitations of this study

► We were able to use a large number of patients (>126 000) aged 50+ years from a diverse range of backgrounds, and a comprehensive list of potential cancer indicators.
► We were able to identify the number of patients presenting to primary care with symptoms, signs, test results or diagnoses, which could potentially indicate cancer, and to explore how this reporting differed by patient characteristics and consultation provision.
► We were unable to take account of whether symptoms were persistent or unexplained, which may inflate our estimates of potential cancer indicators.
► Some symptoms and signs may have been reported in free text rather than using clinical codes or may have been recorded in administration notes not associated with consultations.

## INTRODUCTION

In March 2020, WHO declared COVID-19 a pandemic,[1] and the UK went into lockdown; the public were instructed to 'stay at home, protect the National Health Service and save lives'.[2] Primary care providers were advised to provide all consultations remotely, unless a face-to-face consultation was urgently required,[3] in order to reduce contact and the spread of the virus. Patients at particularly high risk of severe COVID-19 due to age or pre-existing health conditions were advised to 'shield' and avoid all but essential contact.

General practitioner (GP) and nurse consulting rates markedly dropped in April–May 2020 during the period of UK lockdown, but had largely returned to normal by July 2020,[4] with the majority of consultations carried out remotely (mostly via telephone).

There are fears that reduced consulting during the lockdown period and changes in consultation provision may have adversely impacted on cancer detection.[5] There was a

63% reduction in skin cancer diagnosis in England in the first 3 months of the pandemic[6] and 3500 fewer colorectal cancer diagnoses than expected in the first 6 months.[7] This pattern is not limited to the UK. An analysis on the Danish registry showed cancer diagnosis in Denmark dropped by 33% from March to May 2020[8] and similar reductions have been observed in the Netherlands.[9]

These concerning reductions in diagnoses are likely to result from a combination of factors, including temporary suspensions in cancer screening services,[10 11] reduced capacity in secondary care and less reporting of potential cancer symptoms in primary care. A Cancer Research UK survey suggested GPs perceived they received fewer reports of cancer symptoms, particularly from older people, after the pandemic than before.[5] This could be related to changes in the willingness of patients to disclose symptoms, or changes in patient or doctor behaviour associated with the shift to remote consultations. Reduced symptom reporting could result in late diagnoses, increased workload for cancer services and poorer patient outcomes.[12] A modelling study showed that delays in cancer diagnosis in this period could result in approximately 3330 deaths for breast, colorectal, lung and oesophageal cancer alone.[13]

The aim of this study was to investigate how the COVID-19 pandemic affected the number of people aged 50 years and over presenting to primary care with symptoms, signs, test results or diagnoses, which could potentially indicate cancer, and to explore how this reporting differed by patient characteristics and consultation provision (ie, face-to-face vs remote).

## METHODS
### Design and setting
This was a retrospective cohort study of primary care consultations in 21 practices in Bristol, North Somerset and South Gloucestershire, a region in the South West of England. Data were extracted for the RAPCI study (rapid COVID-19 intelligence to improve primary care response), a mixed-methods study on how general practices coped with rapid change to remote consulting in the initial months of the COVID-19 pandemic. The main results are reported elsewhere.[4]

### Data
All 82 practices in Bristol, North Somerset and South Gloucestershire were approached by the Clinical Commissioning Group (CCG) with information about the study. Practices returned expressions of interest to the researchers who selected 21 practices, aiming for a variation of deprivation, ethnic mix, practice size and locality. All participating practices then agreed for anonymised data to be extracted directly from the patient record by One Care, the GP federation in Bristol, North Somerset and South Gloucestershire CCG. One Care provide technical support and management information services to GP practices in the CCG and have the ability to extract

routine data from the patient record via the EMIS enterprise search and report module. All practices use the same EMIS electronic medical records system. Data on all patients registered on 1 July 2020 were included. Patient data included demographics (age, sex, ethnicity and deprivation) and clinical characteristics (such as mental health and shielding status). Data were obtained on all consultations added to the system by clinical staff, and all clinical codes added during those consultations between February 2019 and July 2020 inclusive. We extracted data from February 2019 so that we had a full 18 months to examine trends for the main study. For the analyses in this paper, the period April–July 2020 (ie, the period since the start of UK lockdown) was compared with the same period the previous year (April–July 2019). For this paper, we restricted the analysis to patients aged 50 years and over as cancer is relatively rare in younger age groups.

### Consultations
A consultation was defined as an interaction between a patient and a GP, nurse or paramedic working in general practice. We focused on GP and nurse/paramedic consultations because the number of cancer indicators reported in consultations with other health professionals was very small. We excluded consultations recorded by administrators or other healthcare professionals, and administrative tasks recorded by clinicians. Remote consultations were defined as those carried out by telephone, video or e-consultation. Face-to-face consultations were those carried out in GP practices as well as visits to patients' homes or nursing homes. See online supplemental file 1 for further details.

### Outcomes
Pre-existing code lists[14 15] were used to identify records of potential cancer indicators associated with a consultation. The indicators were collated from the clinical features of undiagnosed cancer (symptoms, signs, abnormal test results or diagnoses) listed in the National Institute for Health and Care Excellence guidance on the recognition and referral of suspected cancer (NG12),[16] using robust methods.[15] Lists were developed using Read codes and subsequently mapped to SNOMED CT codes using medical dictionaries provided by the Clinical Practice Research Datalink for its AURUM dataset.[17] There were a small number of Read codes in the predefined list which do not have equivalent SNOMED CT codes; the frequency with which these codes were used was so low as to be materially insignificant. New SNOMED codes added since the changeover from Read codes to SNOMED did not appear in our predefined lists so we checked this for common indicators cough and back pain, and found new codes were used very rarely.

SNOMED CT codes were used for this analysis. We excluded the indicator for new-onset diabetes (defined as a diabetes code in the absence of any diabetes monitoring in the previous 2 years), because we did not have the necessary patient records required to calculate this.

A binary variable for 'any potential cancer indicators' was derived based on whether a consultation contained at least one potential cancer indicator code (yes) or did not (no). Using these consultation data, the number of patients reporting potential cancer indicators was calculated.

Each potential cancer indicator was categorised based on how many patients reported it in April–July 2019: most commonly (≥0.5%), less commonly (0.1% to <0.5%), rarely (0.02% to <0.1%) and very rarely reported (<0.02%). We felt it was important to separate these, as the most commonly reported indicators include symptoms which often indicate acute illness or chronic illnesses other than cancer (eg, cough), whereas the less commonly/rarely reported indicators are more likely to be associated with cancer (eg, weight loss, lumps and masses).

### Explanatory variables

Age (in July 2020) was split into four categories: 50–59 years, 60–69 years, 70–79 years and 80+ years. Deprivation quintiles (measured by the index of multiple deprivation (IMD) score) were calculated using the IMD deciles recorded in the patient record, which are based on lower super output areas of residence. Ethnicity was derived by mapping ethnicity descriptions in primary care records to one of five ethnicity categories: white, Asian, black, mixed and other (see online supplemental file 1). Presence of a mental health condition was defined as either severe mental illness (defined according to the Quality and Outcomes Framework (QOF) rules[18]), diagnosed depression or prescribed antidepressants (excluding tricyclics, commonly used for non-mental health-related conditions) in the 3 months prior to July 2020. These conditions are part of QOF and are therefore likely to be coded more reliably within GP medical records than other conditions. Sex and shielding status (as of July 2020) were obtained directly from patients' primary care records.

### Statistical analysis

The number and percentage of patients reporting potential cancer indicators during our reporting periods are presented. In addition, consultation rates (and percentages with potential cancer indicators) are reported per 1000 registered patients. Practice list sizes were based on July 2020 data, and adjusted to account for historic list sizes using NHS digital data[19] ('adjusted list size'; see online supplemental file 1).

Changes in proportions of patients presenting (per practice) with any potential cancer indicator in April–July 2020 compared with April–July 2019 were investigated using negative binomial regression models with counts of patients presenting with potential cancer indicators as the outcome; incidence rate ratios (IRRs) and 95% CIs are reported. Consultation year was fitted as a fixed effect, GP practice as a random effect and adjusted practice list size (per level of covariate where appropriate) as the offset. Fixed effects for each of the categorical patient characteristics of interest (age, sex, ethnicity, IMD quintile, mental health status and shielding status), along with an interaction term between each of these covariates and consultation year, were separately fitted to the model; interaction p values are presented and changes in proportions of patients with potential cancer indicators are only presented separately for each level of a covariate if the interaction p value was <0.05. Model validity was checked using standard methods; outliers which disrupted model fit were removed.

To investigate the effect of consultation provision (ie, face-to-face vs remote consulting), we looked at proportions of consultations with potential cancer indicators (out of all consultations) rather than proportions of patients. Separate models were fitted for consultations with GPs and nurses/paramedics. Models were similar to those described above, with number of consultations as the outcome, a fixed effect for consultation provision and an interaction with year, GP practice as a random effect and total numbers of consultations per practice per level of consultation provision as the offset.

For individual potential cancer indicators, unadjusted IRRs for the percentage of patients consulting in April–July 2020 compared with 2019 are presented to help interpretation; 95% CIs are included to indicate uncertainty but, owing to the large number of indicators and resultant issues with multiple testing, no multivariate modelling was performed. Indicators with p<0.0008 have been highlighted. This p value threshold was calculated using the Bonferroni correction, which reduces the standard 0.05 by a factor of the number of tests performed to allow for multiple testing. Within the individual cancer indicators, we highlighted eight alarm symptoms: visible haematuria, rectal bleeding, haematemesis, haemoptysis, jaundice, breast lump, postmenopausal bleeding in women aged 55 years and over and dysphagia. These are symptoms that warrant investigation in their own right because of their predictive power for cancer.[16]

Stata V.15.1 was used to conduct all data checking, cleaning and analyses.

### Patient and public involvement

A patient and public involvement group reviewed the RAPCI proposal before submission, agreed with the importance of the study and commented on the plain language summary. A meeting was also held to discuss preliminary findings of the main study,[4] where a number of points were raised which supported the current work. These included participants highlighting that they felt things were missed during their telephone consultations because the GP could not see them.

### RESULTS

The 21 GP practices recruited covered 25% of Bristol, North Somerset and South Gloucestershire CCG. Selection of practices aimed to be representative of the region

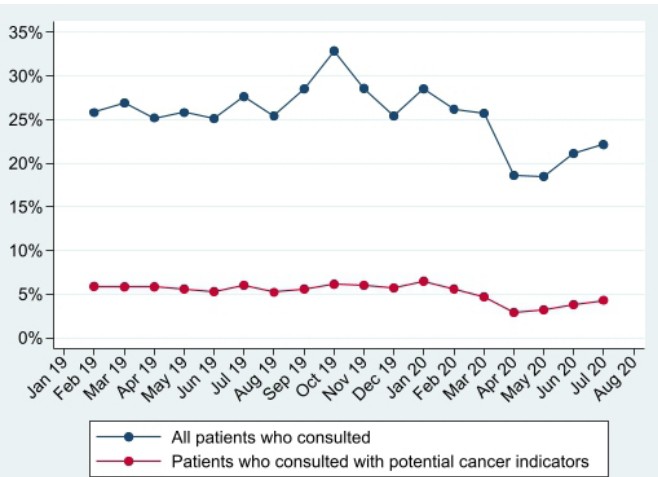

**Figure 1** Proportion of patients consulting and proportion consulting with potential cancer indicators.

in terms of deprivation, size, location and ethnicity. The practices covered 350 966 registered patients (as of July 2020). Of these 123 947 were over 50.

During April–July 2019, of the 123 947 registered patients over 50, 69 254 (56%) consulted, and 21 252 (17%) of patients reported a potential cancer indicator. In April–July 2020, of 126 466 registered patients over 50, 56 314 (45%) consulted and 13 829 (11%) reported an indicator. This comprised 1837 consultations per 1000 patients registered (with indicators reported in 16% of consultations) in April–July 2019 and 1663 consultations per 1000 patients (with indicators reported in 13% of consultations) in April–July 2020. Figures 1 and 2 demonstrate that patients consulting, and consultation rates, substantially dropped in April 2020 (following UK lockdown), and while consultation rates had recovered by July 2020, the proportion of patients who consulted had not.

Furthermore, the percentage of patients reporting potential cancer indicators (figure 1), and percentage of consultations with indicators reported (figure 2), also dropped in April 2020. Both had increased by July

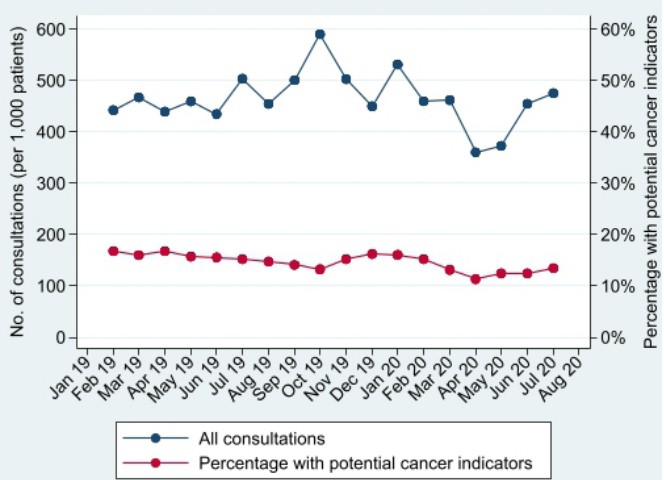

**Figure 2** Number of consultations per 1000 patients and proportion of those with potential cancer indicators.

2020 (to 4.3% of patients and 13.5% of consultations), but neither had recovered to pre-COVID levels (6.0% of patients and 15.2% of consultations in July 2019). Patients who consulted in 2020 were of similar age, sex, IMD quintile and ethnicity as those consulting in 2019, but may have been more likely to have a mental health condition or be shielding (table 1).

There was a 36% reduction in patients who reported potential cancer indicators in April–July 2020 compared with 2019 (IRR 0.64, 95% CI 0.62 to 0.67, p<0.001; table 2). This reduction was stable across age group (interaction p value=0.136), sex (p=0.467), ethnicity (p=0.471), IMD quintile (p=0.585) and shielding status (p=0.099), but differed by mental health status (p<0.001; table 2). The reduction in reporting of these indicators was less pronounced in patients with a mental health condition (IRR 0.73, 95% CI 0.69 to 0.77, p<0.001) than patients without a mental health condition (IRR 0.63, 95% CI 0.61 to 0.66, p<0.001; table 2).

The proportion of consultations where potential cancer indicators were reported differed by consultation provision for both GP and nurse/paramedic consultations (both p<0.001; table 3). In GP consultations, proportions of face-to-face consultations with indicators reported dropped between 2019 and 2020 (IRR 0.84, 95% CI 0.76 to 0.92, p≤0.001) and increased for remote consultations (IRR 1.17, 95% CI 1.07 to 1.29, p=0.001; table 3). For nurse/paramedic consultations, this finding was even more extreme (IRR 0.61, 95% CI 0.44 to 0.83, p=0.002 for face-to-face consultations; IRR 1.60, 95% CI 1.10 to 2.33, p=0.014 for remote consultations). Despite this increase in the proportion of potential cancer indicators in remote consultations, in both years GP remote consultations were less likely to report potential cancer indicators compared with GP face-to-face consultations (table 3).

The proportion of patients reporting all commonly reported indicators reduced between the two periods (unadjusted IRR=0.59), with the biggest reductions in chest infections (IRR=0.25), fever (IRR=0.36), coughs (IRR=0.44) and fatigue (IRR=0.50; table 4). For less commonly reported and rarely reported indicators, the reductions were less marked (IRR=0.75 and 0.76, respectively). Erectile dysfunction (IRR=0.40), irritable bowel syndrome (IRR=0.49), limp/gait abnormalities (IRR=0.39), night sweats (IRR=0.43) and lymphadenopathy (IRR=0.43) saw the largest reductions in these groups. Less common indicators which were similar or even increased were: nausea (IRR=1.00), weight loss (IRR=1.07) and constipation (IRR=1.18). All of the eight alarm symptoms reduced in frequency. Only haematuria (IRR=0.67) saw a significant reduction after correcting for multiple testing (ie, p<0.0008), but at an aggregated level there was a significant reduction in patients who reported at least one cancer alarm symptom (IRR 0.73, 95% CI 0.68 to 0.79, p<0.0008; table 4).

**Table 1** Characteristics of registered and consulting patients in participating practices

| | All patients registered in July 2020 (n=126466)* | | Patients consulting in April–July 2019 (n=69254) | | Patients consulting in April–July 2020 (n=56314) | |
|---|---|---|---|---|---|---|
| | N | % | N | % | N | % |
| **Age (years)** | | | | | | |
| 50–59 | 44614 | 35.3 | 20830 | 30.1 | 16711 | 29.7 |
| 60–69 | 33338 | 26.4 | 17765 | 25.7 | 14252 | 25.3 |
| 70–79 | 27536 | 21.8 | 17512 | 25.3 | 14066 | 25.0 |
| 80+ | 20978 | 16.6 | 13147 | 19.0 | 11285 | 20.0 |
| **Gender** | | | | | | |
| Male | 61798 | 48.9 | 30949 | 44.7 | 25129 | 44.6 |
| Female | 64668 | 51.1 | 38305 | 55.3 | 31185 | 55.4 |
| **Index of multiple deprivation quintile** | | | | | | |
| 1 (most deprived) | 19392 | 15.4 | 10877 | 15.8 | 9355 | 16.7 |
| 2 | 15907 | 12.7 | 8792 | 12.8 | 7403 | 13.2 |
| 3 | 18665 | 14.9 | 10197 | 14.8 | 8151 | 14.5 |
| 4 | 28169 | 22.4 | 15174 | 22.0 | 12236 | 21.80 |
| 5 (least deprived) | 43500 | 34.6 | 23861 | 34.6 | 18888 | 33.7 |
| Missing | 833 | | 353 | | 281 | |
| **Ethnicity†** | | | | | | |
| White | 91401 | 93.2 | 52262 | 93.3 | 42512 | 93.3 |
| Asian | 2507 | 2.6 | 1443 | 2.6 | 1161 | 2.5 |
| Black | 2963 | 3.0 | 1655 | 3.0 | 1374 | 3.0 |
| Mixed | 1000 | 1.0 | 532 | 0.9 | 425 | 0.9 |
| Other | 227 | 0.2 | 111 | 0.2 | 89 | 0.2 |
| Missing | 28368 | | 13251 | | 10753 | |
| **MH status** | | | | | | |
| No MH conditions | 111143 | 87.9 | 57923 | 83.6 | 45885 | 81.5 |
| 1+ MH condition | 15323 | 12.1 | 11331 | 16.4 | 10429 | 18.5 |
| **Shielding status** | | | | | | |
| Not advised to shield | 116518 | 92.1 | 61417 | 88.7 | 48305 | 85.8 |
| Advised to shield | 9948 | 7.9 | 7837 | 11.3 | 8009 | 14.2 |

*The corresponding number of patients registered in 2019 was 123947. As we only had patient characteristic data for the 21 participating practices for patients registered on 1 July 2020, we imputed the 2019 list sizes using NHS digital historical list sizes.
†White includes British, white British, mixed British and other white background. Black includes African, Caribbean, Somali and black British. Asian includes Indian, British Indian, Pakistani, Chinese and other Asian background. Mixed includes white and black Caribbean, white and Asian, white and African, other mixed white and other mixed background. Other includes Turkish, Arab, Iranian, other ethnic non-mixed and any other group. See online supplemental file 1 for further details.
MH, mental health; NHS, National Health Service.

## DISCUSSION
### Summary of findings
In April 2020, coinciding with UK lockdown, both the proportion of patients consulting, and those presenting with features potentially indicating cancer reduced compared with April 2019. By July 2020, although overall consultation rates had returned to previous levels, the proportion of patients consulting, and the proportion with potential cancer indicators remained lower than the previous year. The reduction in potential cancer indicator reporting did not differ by age, sex, ethnicity, deprivation or shielding status, but was less evident in patients with mental health conditions than those with no mental health conditions. Our results show that, despite an increase in consultations (for any reason) in patients aged 85+ years and patients advised to shield (shown in our previous paper[4]), these have not resulted in higher levels of potential cancer indicator reporting in these groups, perhaps indicating these consultations were for routine assessment rather than to address new issues.

**Table 2** Patients with any potential cancer indicator reported in April–July 2020 compared with 2019

| | April–July 2019 (n=123947) | | April–July 2020 (n=126466) | | Change in proportion of patients with potential cancer indicators (2020 vs 2019) | | |
| --- | --- | --- | --- | --- | --- | --- | --- |
| | N | %* | N | %* | IRR | 95% CI | P value |
| Overall | 21252 | 17 | 13829 | 11 | 0.64 | 0.62 to 0.67 | <0.001 |
| **Age (years)** | | | | | | | 0.136† |
| 50–59 | 6217 | 14 | 4083 | 9 | 0.65 | 0.61 to 0.69 | <0.001 |
| 60–69 | 5398 | 17 | 3368 | 10 | 0.61 | 0.58 to 0.65 | <0.001 |
| 70–79 | 5457 | 20 | 3449 | 13 | 0.62 | 0.59 to 0.67 | <0.001 |
| 80+ | 4180 | 20 | 2929 | 14 | 0.68 | 0.64 to 0.73 | <0.001 |
| **By sex** | | | | | | | 0.467† |
| Male | 9202 | 15 | 5905 | 10 | 0.63 | 0.60 to 0.67 | <0.001 |
| Female | 12050 | 19 | 7924 | 12 | 0.65 | 0.62 to 0.68 | <0.001 |
| **By ethnicity ‡** | | | | | | | 0.471† |
| White | 15910 | 18 | 10340 | 11 | 0.64 | 0.61 to 0.68 | <0.001 |
| Asian | 590 | 24 | 357 | 14 | 0.58 | 0.50 to 0.69 | <0.001 |
| Black | 710 | 24 | 497 | 17 | 0.68 | 0.58 to 0.80 | <0.001 |
| Mixed | 193 | 20 | 106 | 11 | 0.55 | 0.42 to 0.70 | <0.001 |
| Other | 40 | 18 | 25 | 11 | 0.60 | 0.36 to 1.00 | 0.050 |
| **By index of multiple deprivation quintile** | | | | | | | 0.585† |
| 1 (most deprived) | 3916 | 21 | 2662 | 14 | 0.67 | 0.63 to 0.72 | <0.001 |
| 2 | 2978 | 19 | 1981 | 12 | 0.64 | 0.60 to 0.69 | <0.001 |
| 3 | 3158 | 17 | 2052 | 11 | 0.63 | 0.59 to 0.68 | <0.001 |
| 4 | 4607 | 17 | 2999 | 11 | 0.63 | 0.59 to 0.67 | <0.001 |
| 5 (least deprived) | 6469 | 15 | 4053 | 9 | 0.62 | 0.58 to 0.67 | <0.001 |
| **By MH status** | | | | | | | <0.001† |
| No MH conditions | 17338 | 16 | 10924 | 10 | 0.62 | 0.60 to 0.64 | <0.001 |
| 1+ MH condition | 3914 | 26 | 2905 | 19 | 0.73 | 0.69 to 0.77 | <0.001 |
| **By shielding status** | | | | | | | 0.099† |
| Not advised to shield | 17956 | 16 | 11564 | 10 | 0.63 | 0.61 to 0.66 | <0.001 |
| Advised to shield | 3296 | 34 | 2265 | 23 | 0.68 | 0.63 to 0.72 | <0.001 |

*Percentages are calculated out of all patients in each category.
†P values presented on the blue rows are the p values for the interaction between year and the given characteristic; IRR only presented by each level of a characteristic if this p<0.05.
‡White includes British, white British, mixed British and other white background. Black includes African, Caribbean, Somali and black British. Asian includes Indian, British Indian, Pakistani, Chinese and other Asian background. Mixed includes white and black Caribbean, white and Asian, white and African, other mixed white and other mixed background. Other includes Turkish, Arab, Iranian, other ethnic non-mixed and any other group. See online supplemental file 1 for further details.
IRR, incidence rate ratio; MH, mental health.

Patients with mental health conditions did, however, have a smaller reduction in presentations with features potentially indicating cancer than other patients, perhaps as a result of this increased focus on them from GPs.[4] Between 2019 and 2020, there was a large reduction in face-to-face consultations which involved reporting of cancer indicators, which was not fully compensated for by an increase in reporting of these indicators in remote consultations. Of the most commonly reported indicators, chest infections, fever, coughs and fatigue reduced most dramatically. The reduction in 'alarm' indicators (27%) was less pronounced than the overall reduction (36%) but was still significant (p<0.0008).

**Strengths and limitations**

To our knowledge, this is the first paper to assess the effect of the COVID-19 pandemic on potential cancer indicator reporting in UK primary care. The analysis is based on a comprehensive list of potential cancer indicators, developed and validated by price, which has been used in previous research and is available on request.[14 15] Our analysis included a large number of patients (>126000) aged 50+ years from a

**Table 3** Consultations (per 1000 patients) with potential cancer indicators reported in April–July 2020 compared with 2019

| | April–July 2019 | | April–July 2020 | | Change in rates of consultations with potential cancer indicators (2020 vs 2019) | | |
|---|---|---|---|---|---|---|---|
| **GP consultations** | **N** | **%*** | **N** | **%*** | **IRR** | **95% CI** | **P value** |
| Overall | 236.5 | 19 | 176.4 | 15 | 0.79 | 0.74 to 0.84 | <0.001 |
| By consultation provision | | | | | | | <0.001† |
| Face-to-face | 187.8 | 21 | 25.4 | 17 | 0.84 | 0.76 to 0.92 | <0.001 |
| Remote | 48.7 | 13 | 151.0 | 14 | 1.17 | 1.07 to 1.29 | 0.001 |
| **Nurse/Paramedic consultations** | **N** | **%** | **N** | **%** | **IRR** | **95% CI** | **P value** |
| Overall | 53.6 | 9 | 31.9 | 7 | 0.69 | 0.56 to 0.85 | <0.001 |
| By consultation provision | | | | | | | <0.001† |
| Face-to-face | 50.2 | 10 | 15.7 | 6 | 0.61 | 0.44 to 0.83 | 0.002 |
| Remote | 3.4 | 7 | 16.1 | 9 | 1.60 | 1.10 to 2.33 | 0.014 |

IRRs and corresponding 95% CIs and p values from negative binomial models.

*Percentages are calculated out of all consultations in each category.

†P values presented on these rows are the p values for the interaction between year and consultation provision.

GP, general practitioner; IRR, incidence rate ratio.

diverse range of backgrounds, and as changes in service provision due to the pandemic have affected the whole country, our findings are likely to be generalisable to other areas of England. As all patients were included, there was no selection bias within practices. We endeavoured to select practices that were representative of the region but, as participation was voluntary, there may have been some selection bias at a practice level. However, this is unlikely to have changed the conclusions regarding reductions in potential cancer indicators, as there is no reason to suppose research active practices would have a greater reduction than other practices. The representativeness of the data is best demonstrated with reference to the full RAPCI dataset (350 966 patients) not just the sample of 50+ patients in this dataset. The age and gender profile of the RAPCI data are representative of the UK. The proportion of ethnic minorities in the RAPCI data (12%) is representative of the CCG, but slightly less than the UK as a whole.[20] The deprivation mix was slightly polarised; a fully representative sample would have approximately 20% in each quintile. The full RAPCI dataset had 20% in the bottom and fourth quintile but 27% in the most affluent so the second and third quintiles were slightly under-represented. Except for ethnicity, missing data about patient characteristics were low. Twenty-six per cent of ethnicity data was missing in the RAPCI dataset (22% in the over 50s subset used for this paper). This may have led to information bias.

There are several limitations which pertain to the recording of potential cancer indicators. First, some symptoms and signs may have been reported in free text rather than using clinical codes,[21] or may have been recorded in administration notes not associated with consultations. Furthermore, as described in the 'Methods' section, there were a small number of Read codes in the predefined list which do not have equivalent SNOMED CT codes, although the frequency with which these codes were used was so low as to be materially insignificant. These issues may have resulted in systematic under-reporting of potential cancer indicators; however, this is likely to be similar in both years and therefore should not affect the comparative findings. A further limitation was that new SNOMED codes added since the changeover from Read codes to SNOMED did not appear in our predefined lists, an issue which would affect April–July 2020 but not 2019; however, in checking the common indicators cough and back pain, we found new codes were used very rarely (data not shown). Finally, for indicators to be of concern as possible harbingers of cancer, they are sometimes qualified by age at onset, or as persistent or unexplained in National Institute for Health and Care Excellence suspected cancer guidance (NG12). Therefore, we limited our analysis to patients over 50 years of age (because cancer is more common in older patients) but were unable to take account of whether symptoms were persistent or unexplained. This limitation applies to other studies based on routine records and using similar methods,[14] but the effect may be to inflate our estimates of potential cancer indicators.

### Comparison with other literature

Few other studies have looked at the content of primary care consultations following UK lockdown during the COVID-19 pandemic. Analysis of primary care data from a deprived urban population found that diagnoses of common conditions decreased substantially between March and May 2020, suggesting a large number of patients may have undiagnosed conditions resulting from changes in access post-lockdown.[22] Our study adds to these findings for cancer-specific diagnoses.

Cancer screening programmes, estimated to account for about 5% of cancer diagnoses, were suspended in the wake of the pandemic.[10] This made symptom-based diagnosis all the more urgent. UK lockdown could have affected symptom-based reporting via the move to remote consulting. Research has shown that telephone and video consultations result in

**Table 4** Patients with individual potential cancer indicator reported in April–July 2020 compared with 2019

| | April–July 2019 (n=123 947) | | April–July 2020 (n=126 466) | | | 95% CI (bold indicates p<0.0008) |
|---|---|---|---|---|---|---|
| | N | % | N | % | IRR | |
| Any potential cancer indicator | 21 252 | 17.15 | 13 829 | 10.93 | 0.64 | **0.62 to 0.65** |
| **Cancer alarm symptoms (see asterisk (*))** | 1801 | 1.45 | 1344 | 1.06 | 0.73 | **0.68 to 0.79** |
| | | | | | | |
| **Most commonly reported indicators (≥0.5% of patients in 2019)** | | | | | | |
| *Any commonly reported indicator* | *16 632* | *13.42* | *10 054* | *7.95* | *0.59* | **0.58 to 0.61** |
| Chest infection | 1427 | 1.15 | 370 | 0.29 | 0.25 | **0.23 to 0.29** |
| Fever | 1216 | 0.98 | 443 | 0.35 | 0.36 | **0.32 to 0.40** |
| Cough | 3125 | 2.52 | 1397 | 1.10 | 0.44 | **0.41 to 0.47** |
| Fatigue | 1284 | 1.04 | 655 | 0.52 | 0.50 | **0.45 to 0.55** |
| Urinary tract infection | 760 | 0.61 | 394 | 0.31 | 0.51 | **0.45 to 0.57** |
| Pain in shoulder | 1307 | 1.05 | 785 | 0.62 | 0.59 | **0.54 to 0.64** |
| Proteinuria | 697 | 0.56 | 437 | 0.35 | 0.61 | **0.54 to 0.69** |
| Non-visible haematuria | 979 | 0.79 | 616 | 0.49 | 0.62 | **0.56 to 0.68** |
| Pain in abdomen | 1908 | 1.54 | 1301 | 1.03 | 0.67 | **0.62 to 0.72** |
| *Haematuria** | 897 | 0.72 | 612 | 0.48 | 0.67 | **0.60 to 0.74** |
| Shortness of breath | 2382 | 1.92 | 1650 | 1.30 | 0.68 | **0.64 to 0.72** |
| Pain in chest | 1180 | 0.95 | 830 | 0.66 | 0.69 | **0.63 to 0.75** |
| Back pain/Backache | 2659 | 2.15 | 1887 | 1.49 | 0.70 | **0.66 to 0.74** |
| Lower urinary tract symptoms | 1402 | 1.13 | 1027 | 0.81 | 0.72 | **0.66 to 0.78** |
| Compressed trachea | 864 | 0.70 | 638 | 0.50 | 0.72 | **0.65 to 0.80** |
| Diarrhoea | 860 | 0.69 | 645 | 0.51 | 0.74 | **0.66 to 0.82** |
| **Less commonly reported indicators (0.1% to <0.5% of patients in 2019)** | | | | | | |
| *Any less commonly reported indicator* | *6585* | *5.31* | *5008* | *3.96* | *0.75* | **0.72 to 0.77** |
| Erectile dysfunction | 363 | 0.29 | 148 | 0.12 | 0.40 | **0.33 to 0.49** |
| Irritable bowel syndrome | 194 | 0.16 | 97 | 0.08 | 0.49 | **0.38 to 0.63** |
| Gynae abnormalities (eg, menstrual disorders and unusual bleeding) | 190 | 0.15 | 99 | 0.08 | 0.51 | **0.40 to 0.65** |
| Pain in throat (eg, tonsilitis) | 201 | 0.16 | 109 | 0.09 | 0.53 | **0.42 to 0.67** |
| Iron deficiency anaemias | 477 | 0.38 | 273 | 0.22 | 0.56 | **0.48 to 0.65** |
| Change in bowel habit | 281 | 0.23 | 161 | 0.13 | 0.56 | **0.46 to 0.68** |
| Pain in breast | 170 | 0.14 | 99 | 0.08 | 0.57 | **0.44 to 0.74** |
| Bruising | 208 | 0.17 | 126 | 0.10 | 0.59 | **0.47 to 0.74** |
| Prostate-specific antigen raised/abnormal | 173 | 0.14 | 121 | 0.10 | 0.69 | **0.54 to 0.87** |
| *Postmenopausal bleeding** | 144 | 0.12 | 101 | 0.08 | 0.69 | 0.53 to 0.89 |
| Pain in pelvis | 388 | 0.31 | 280 | 0.22 | 0.71 | **0.60 to 0.83** |
| Pruritus | 356 | 0.29 | 259 | 0.20 | 0.71 | **0.61 to 0.84** |
| Vomiting | 245 | 0.20 | 180 | 0.14 | 0.72 | **0.59 to 0.88** |
| Dyspepsia | 590 | 0.48 | 450 | 0.36 | 0.75 | **0.66 to 0.85** |
| Distension of abdomen | 142 | 0.11 | 111 | 0.09 | 0.77 | 0.59 to 0.99 |
| Reflux | 560 | 0.45 | 439 | 0.35 | 0.77 | **0.68 to 0.87** |
| *Lump in breast** | 208 | 0.17 | 170 | 0.13 | 0.80 | 0.65 to 0.99 |
| Recurring infections | 366 | 0.30 | 304 | 0.24 | 0.81 | 0.70 to 0.95 |
| *Rectal bleeding** | 357 | 0.29 | 303 | 0.24 | 0.83 | 0.71 to 0.97 |

Continued

**Table 4**   Continued

| | April–July 2019 (n=123 947) | | April–July 2020 (n=126 466) | | | 95% CI (bold indicates p<0.0008) |
|---|---|---|---|---|---|---|
| | N | % | N | % | IRR | |
| Dysuria | 196 | 0.16 | 169 | 0.13 | 0.85 | 0.68 to 1.04 |
| Vaginal discharge | 153 | 0.12 | 134 | 0.11 | 0.86 | 0.68 to 1.09 |
| *Dysphagia** | 171 | 0.14 | 151 | 0.12 | 0.87 | 0.69 to 1.08 |
| Lump (excluding breast) | 402 | 0.32 | 375 | 0.30 | 0.91 | 0.79 to 1.06 |
| Nausea | 147 | 0.12 | 150 | 0.12 | 1.00 | 0.79 to 1.26 |
| Weight loss | 218 | 0.18 | 239 | 0.19 | 1.07 | 0.89 to 1.30 |
| Constipation | 545 | 0.44 | 655 | 0.52 | 1.18 | 1.05 to 1.32 |
| **Rarely reported indicators (0.02% to <0.1% of patients in 2019)** | | | | | | |
| *Any rarely reported indicator* | 909 | 0.73 | 702 | 0.56 | 0.76 | **0.68 to 0.84** |
| Limp/Gait abnormalities | 25 | 0.02 | 10 | 0.01 | 0.39 | 0.17 to 0.85 |
| Night sweats | 75 | 0.06 | 33 | 0.03 | 0.43 | **0.28 to 0.66** |
| Lymphadenopathy | 86 | 0.07 | 38 | 0.03 | 0.43 | **0.29 to 0.64** |
| *Haemoptysis** | 62 | 0.05 | 34 | 0.03 | 0.54 | 0.34 to 0.83 |
| Abnormal chest X-ray | 28 | 0.02 | 17 | 0.01 | 0.60 | 0.31 to 1.13 |
| Hoarseness of voice | 101 | 0.08 | 65 | 0.05 | 0.63 | 0.45 to 0.87 |
| Pain in testis | 71 | 0.06 | 46 | 0.04 | 0.63 | 0.43 to 0.93 |
| Pain in bones | 118 | 0.10 | 86 | 0.07 | 0.71 | 0.53 to 0.95 |
| Possible bone and soft tissue sarcoma symptoms (eg, abnormal X-ray) | 30 | 0.02 | 22 | 0.02 | 0.72 | 0.40 to 1.29 |
| Chest/Breathing abnormalities | 26 | 0.02 | 20 | 0.02 | 0.75 | 0.40 to 1.40 |
| Mass in neck | 115 | 0.09 | 89 | 0.07 | 0.76 | 0.57 to 1.01 |
| Mass in testis | 102 | 0.08 | 79 | 0.06 | 0.76 | 0.56 to 1.03 |
| Superior vena cava syndrome | 39 | 0.03 | 38 | 0.03 | 0.95 | 0.59 to 1.53 |
| Testis swelling/Abnormalities | 31 | 0.03 | 32 | 0.03 | 1.01 | 0.60 to 1.71 |
| Appetite loss | 49 | 0.04 | 76 | 0.06 | 1.52 | 1.05 to 2.22 |
| Mass in abdomen | 42 | 0.03 | 79 | 0.06 | 1.84 | **1.25 to 2.75** |
| **Very rarely reported indicators (<0.02% of patients in 2019)** | | | | | | |
| *Any very rarely reported indicator†* | 141 | 0.11 | 108 | 0.09 | 0.75 | 0.58 to 0.97 |

Bold face indicates a change over time based on 95% CI adjusted for multiple testing using the Bonferroni correction (P<0.0008).

*'Alarm' symptoms, which warrant investigation in their own right in NICE suspected cancer guidelines, because of their predictive power for cancer: visible haematuria, rectal bleeding, haematemesis, haemoptysis, jaundice, breast lump, postmenopausal bleeding in women aged 55 years and over and dysphagia.

†Very rare indicators affecting <0.02% of patients (25 individuals) in April–July 2019 clubbing of fingers, ulceration of vulva, prostate abnormalities on digital rectal exam, nipple retraction, skin lesion, mass in thyroid, jaundice, spinal cord compression, mass in pelvis, faecal occult blood test, abnormal skin on breast, nipple discharge, haematemesis and 12 additional indicators with incidence <5 patients in April–July 2019.

‡Indicators are ordered from largest to smallest reduction (by IRR) within each commonality grouping. These figures only include indicators coded in routinely collected records using SNOMED codes. In some cases, GPs/nurses may enter symptoms in free text, and/or only code their diagnosis, which may lead to under-reporting here; this under-recording is likely to be similar in both years.

IRR, incidence rate ratio; NICE, National Institute for Health and Care Excellence.

fewer problems being identified than face-to-face consultations.[23] Lockdown could also have influenced patients to only contact primary care if they thought their problem was serious, which is likely to be reflected in the reduced proportion of patients consulting found in our study.

Early in the pandemic, GPs predicted that patients with well-recognised red-flag symptoms, such as a new lump or rectal bleeding, would continue to present to primary care, but that vaguer cancer symptoms such as fatigue, change in bowel habit and weight loss might be dismissed by the patient as trivial and not presented to primary care.[10] This was supported by our findings, which show that the most common symptoms reduced more substantially than less common ones. Of course, many of these more common

symptoms (such as chest infections and sore throats) will be attributable to causes other than cancer, such as viral infections, and the figures should be interpreted in this context. Social distancing and lockdown has resulted in less infection overall as demonstrated by the Royal College of General Practitioners research and surveillance centre data, which shows a marked reduction in weekly reported incidence of asthma, intestinal infectious diseases and upper and acute respiratory tract infections from week 12 of 2020.[24] Nonetheless, the reductions are still concerning, especially for older patients. A 2020 survey from Cancer Research UK suggested GPs are particularly worried about older people not consulting with potential cancer symptoms compared with before the pandemic.[5] Analysis of the national cancer registry in the Netherlands shows reduction in cancer diagnosis at the start of the pandemic was particularly pronounced in patients aged 80 years or over.[9] The Netherlands operates a similar GP gatekeeper model to the UK,[25] and the authors of this Dutch paper suggested that the move to remote GP consulting combined with patients delaying consultation about potential non-specific cancer symptoms may have contributed to the reduction in diagnoses.[9] In the current study, we show that this suggestion is supported for the set of UK practices we analysed. Although we did not observe a greater reduction in symptom reporting in patients aged 80 years or over, potential symptoms in over 80s are more likely to result in a cancer diagnosis; therefore, the fact that reductions were similar across all age groups, rather than less of a reduction seen in older age groups, is of concern.

## Implications for policy and practice

Combined with our previous work,[4] the Cancer Research UK report,[5] and reductions in cancer diagnoses,[6 7] these findings suggest that patients are less likely to report potential cancer indicators than before the COVID-19 pandemic, particularly for more common symptoms such as fever and coughs and for alarm symptoms. In the context of repeated lockdowns, it is therefore extremely important that the general public are advised that they should still consult with primary care if they have persistent symptoms that might indicate cancer. Furthermore, GPs and nurses should be encouraged to ask more probing questions during remote consulting, as they may miss symptoms which may have previously been picked up from non-verbal cues and possibly a more open discussion face-to-face.

## CONCLUSION

The proportion of patients consulting with primary care, as well as those reporting potential cancer indicators, reduced during the first wave of the COVID-19 pandemic. While there has been an increased rate of potential cancer indicators reported in remote consultations compared with last year, there are still lower rates in remote GP consultations than in face-to-face, suggesting remote consulting may be part of the reason for the reduction in reporting of potential cancer indicators.

**Acknowledgements** The authors would like to thank all the participants in this study, Bristol North Somerset and South Gloucestershire Clinical Commissioning Group, One Care for providing the data extract, the NIHR Clinical Research Network for adopting the study on the NIHR portfolio and the NIHR SPCR for funding the research. The authors would also like to thank Willie Hamilton for his work on developing the potential cancer indicator lists (alongside author Sarah Price).

**Contributors** LJS, MM, RD, JH and CS contributed to the conception and design of the study, SP provided the potential cancer indicator code lists and interpretation of these, RL extracted the data, MM led the main study and LJS performed the analysis and drafted the manuscript. MM updated the paper following reviewer comments and TP performed analysis for this. All authors contributed to the organisation and conduct of the study, the interpretation of study data and results and critiqued the manuscript for important intellectual content.

**Funding** This study was funded by the National Institute for Health Research (NIHR) School for Primary Care Research. Additional funding for staff time was provided by the Applied Research Collaboration West (ARC West) at University Hospitals Bristol and Weston NHS Foundation Trust and One Care. CS is a NIHR Senior Investigator. SP is funded by the NIHR Policy Research Programme via the Policy Research Unit in Cancer Awareness, Screening and Early Diagnosis, PR-PRU-1217-21601. TP is supported by the Integrative Epidemiology Unit, which receives funding from the UK Medical Research Council and the University of Bristol (MC_UU_00011/1 and MC_UU_00011/3).

**Disclaimer** The funder had no role in the study design, collection and analysis of data, or the writing of the manuscript. The views expressed are those of the authors and not necessarily those of, the NIHR or the Department of Health and Social Care.

**Competing interests** None declared.

**Patient consent for publication** Not required.

**Ethics approval** This study received ethical approval from the University of Bristol Faculty of Health Sciences Research Ethics Committee (ID 103166), and Health Research Authority approval (IRAS project ID 282541; REC reference 20/HRA/2070). The study was sponsored by the University of Bristol.

**Provenance and peer review** Not commissioned; externally peer reviewed.

**Data availability statement** Data are available on reasonable request. Under the Data Transfer Standard Operating Procedure agreed with practices, the patient data for this study will not be shared outside the research team. The codes lists used in the analysis which identify symptoms, signs, test results or diagnoses which could potentially indicate cancer can be made available on reasonable request.

**ORCID iDs**
Lauren J Scott http://orcid.org/0000-0003-3129-5123
Mairead Murphy http://orcid.org/0000-0002-3550-2727
Tom Palmer http://orcid.org/0000-0003-4655-4511
Chris Salisbury http://orcid.org/0000-0002-4378-3960

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
