## [Reviewer comments · BMJ Open]

ARTICLE DETAILS

TITLE (PROVISIONAL)	Changes in presentations with features potentially indicating cancer in primary care during the COVID-19 pandemic: a retrospective cohort study
AUTHORS	Scott, Lauren; Murphy, Mairead; Price, Sarah; Lewis, Rhys; Denholm, Rachel; Horwood, Jeremy; Palmer, Tom; Salisbury, Chris

VERSION 1 – REVIEW

REVIEWER	Jones, Daniel University of Leeds
REVIEW RETURNED	25-Feb-2021

GENERAL COMMENTS	Thank you for asking me to review this interesting retrospective cohort study on the changes in presentation of cancer symptoms to primary care during the COVID 19 pandemic. I have enjoyed reading the paper and feel it would be a helpful addition to the literature. Abstract: Generally clear, giving a fair and balanced report of the study. There are some long sentences which make the abstract harder to follow than perhaps it needs to be. Introduction: Very clear, well written, highlights the importance of the study. Referenced well. Method: Detailed description of the methods which would allow for repetition of the study. I was left wondering why just GP, nurse and paramedic consultations were included especially with the rise of other allied health professionals such as pharmacists and physician associates. I also wondered if the way in which the presence of a mental health condition was adequate? Is being on the severe mental illness register (probably a very small number) or having depression adequate? What about anxiety / personality disorder and the multiple other mental health conditions which could affect presentation? It was good to see the input of a PPI group on the study. Results: Well written and clear Discussion: The discussion is generally clear and well written.
---

	I wonder if the problems with the SNOMED codes should be mentioned in the methods. I agree with the authors that it is unlikely to have affected the results. I am left wondering, and probably feel there should be some comment on the role of social distancing and lockdown on the fact there has probably been less actual chest infections this year and as a result probably less fever / SOB and arguably fatigue. Is this because patients are not reporting cancer symptoms? Or is this because genuinely there has been less infections? We know for example the recorded number of flu cases this year is dramatically lower than usual. You could maybe argue that the number of cases of shoulder pain / back pain / chest pain have reduced because fewer people are working / exercising and therefore injuring themselves, and therefore do not have symptoms rather than not reporting symptoms they are having.
--	--

REVIEWER	Virgilsen, Line Aarhus Universitet
REVIEW RETURNED	02-Mar-2021

GENERAL COMMENTS	An interesting and important work which shed light on the potential side effects of pandemic control measures such as lock-downs. The paper take advantage of a large sample with information from a representative area of the UK. This review suggest some elaborations to the methods and results. Abstract: In the design, setting and participants you could include the sample size of included patients Result section: As the method (in the abstract) do not explain that you are comparing two periods, I would suggest to make this clear. E.g. by beginning your result section with: "Comparing April-July 2019 with the same period in 2020,". Also, I would skip including the numbers (69,254/123947 etc.) which will draw more attention to the numbers of particular interest for this study: 17% in 2019 versus 11% in 2020 which presented with potential cancer symptoms. Method section: Design and setting: Could the authors include the number of patients what the included 21 practices are covering? If relevant, also the proportion of all GPs in South West of England and how these 21 practices were selected for the RAPCI study – does they represent well the national population of patients (maybe cover this in discussion if the authors find it more appropriate)? Data: I think the data collection and method could be explained a bit more clear. First of all, could the authors elaborate a bit about the origin of the data source? Data must have been initiated prior to the COVID since the authors can compare to 2019 data. This is such a strength and I would like to understand this a bit better when reading. Second; the data selection for this paper could be explained a bit more clear. When reading it for the first time, I was confused by the different period mentioned, and I had to read it a few times to be clear when you were referring to the selection of patients (patients alive and registered with a practice 1th of July
--

2020) and time period included (all registered contacts from Feb 2019 until 31 July 2020). Yet, why was February 2019 included when the comparison period was from April?

Results:

How did the authors get the 56%? From table 1, $69,254/126,466=54.8\%$ (55%). Maybe a typo? I would suggest that the authors include a total column in table 1 which would include information about the proportion of patients consulted among all registered patients per 1 July 2020. If this is included, you could skip these calculations in the result section.

Page 6 line 3-7: the authors state that the levels of consultations did not recover to prior COVID levels. Could the authors refer to a table/figure where this can clearly be assessed? (figure 2? – maybe state the proportions in the text (7% vs 4%??))

Page 6 line 31 - /table 3: I acknowledge your point with multiple testing for each symptom/condition. But for me, I would still prefer that you e.g. marked in bold the IRR with a statistically significant reduction from 2019 to 2020. Especially among the rarer symptoms, with a very low incidence, this is relevant to assess the impact. Maybe consider using the Bonferroni correction to correct for multiple testing.

Further, in relation to table 3 and the important issue raised by the authors in the discussion concerning the red-flag symptoms. I do acknowledge the authors choice in dividing the symptoms into “commonly/not-commonly reported symptoms”. But I wonder; in terms of cancer diagnostics, if it could bring important information to divide the symptoms according to alarm-symptoms: yes no. I would expect patients to be more likely to present an alarm symptom in a remote consultation than would be the case if the patient experienced a vague or unspecific symptom. This has great implications as the vague and unspecific symptoms often is linked with “harder to diagnose” cancer types, which also have inferior cancer prognosis in general and could be suitable to a further survival disparities if these cancers are less likely to be caught during a lock down. The current categorisations of symptoms does not offer this distinction. If possible, the authors could add an analysis comparing the likelihood of presenting alarm symptom vs non-alarm symptoms during the lock-down period.

Page 8: you refer to “in the over 80s” – this could be more clear throughout the section: “patients aged 80 years or older” (e.g.).

Table 1: besides including a “total column” as mentioned, could you consider changing “all patients registered in July 2020” into “all patients registered per 1 July 2020”

Figure 2: Please correct the typo (‘000 should be 1000)

Discussion: Bias: I think the strengths and limitation section could be strengthened by referring to selection bias and information bias. The authors do to some extent cover this (You do mention that a large proportion of patients with diverse backgrounds was included, but is there reason to suspect selection bias among the selected 21 practices compared to the background practices of your study population – all practices of the UK?). You also mention the risk of information bias – without referring to is as such.

	Other than this; congratulations on your work and good luck.
--	--

VERSION 1 – AUTHOR RESPONSE

Reviewer: 1

	Section	Reviewer comment	Author response
2	Abstract	There are some long sentences which make the abstract harder to follow than perhaps it needs to be.	We have adjusted the abstract, making the language more succinct so that it is easier to read. This overlaps with point 7 (reviewer 2) which gave some specific suggestions to make the abstract clearer and we have also implemented these as well as making the language clearer.
3	Methods	I was left wondering why just GP, nurse and paramedic consultations were included especially with the rise of other allied health professionals such as pharmacists and physician associates	This is because the number of consultations with other health professionals was proportionally very small. We therefore decided to make GPs and nurse/paramedics the focus of the paper. We have added this explanation to the paper.
4	Methods	I also wondered if the way in which the presence of a mental health condition was adequate? Is being on the severe mental illness register (probably a very small number) or having depression adequate? What about anxiety / personality disorder and the multiple other mental health conditions which could affect presentation?	We included severe mental illness and depression within our explanatory variables because these conditions are included in the Quality and Outcomes Framework and are therefore likely to be coded more reliably within GP medical records than other conditions.
5	Discussion	I wonder if the problems with the SNOMED codes should be mentioned in the methods. I agree with the authors that it is unlikely to have affected the results.	We have now added this to the methods.
6	Discussion	I am left wondering, and probably feel there should be some comment on the role of social distancing and lockdown on the fact there has probably been less actual chest infections this year and as a result probably less fever / SOB and arguably fatigue. Is this because patients are not reporting cancer symptoms? Or is this because genuinely there has been less infections? We know for example the recorded number of flu cases this year is dramatically lower than usual. You could	We had added a comment on this already in terms of less infections spreading. We have now expanded this to make an explicit link with social distancing. As the reviewer points out, there is clear evidence that social distancing probably resulted in less infection spreading. However, we would

	Section	Reviewer comment	Author response
		maybe argue that the number of cases of shoulder pain / back pain / chest pain have reduced because fewer people are working / exercising and therefore injuring themselves, and therefore do not have symptoms rather than not reporting symptoms they are having.	prefer not to add the comment about shoulder and back pain, as it is more speculative. Many people in manual jobs that might result in injury continued working through lockdown. Furthermore, many office workers were working at home from laptops with less access to ergonomically designed chairs and desk spaces than their employers would normally provide. This could have resulted in increased back pain.

Reviewer: 2

	Section	Reviewer comment	Author response
7	Abstract	In the design, setting and participants you could include the sample size of included patients	We have added this as suggested.
8	Abstract (Results section)	As the method (in the abstract) do not explain that you are comparing two periods, I would suggest to make this clear. E.g. by beginning your result section with: "Comparing April-July 2019 with the same period in 2020,". Also, I would skip including the numbers (69,254/123947 etc.) which will draw more attention to the numbers of particular interest for this study: 17% in 2019 versus 11% in 2020 which presented with potential cancer symptoms.	We have adjusted the abstract now to make this clear. It now says: "During April-July 2019, 17% of registered patients aged 50+ years reported a potential cancer indicator in a consultation with a GP or nurse. During April-July 2020 this reduced to 11%."
9	Methods	Could the authors include the number of patients what the included 21 practices are covering? If relevant, also the proportion of all GPs in South West of England and how these 21 practices were selected for the RAPCI study – does they represent well the national population of patients (maybe cover this in discussion if the authors find it more appropriate)?	We have added this now. I have added the proportion of GP practices in Bristol, North Somerset and South Gloucestershire CCG (not South West England), as the practices were only selected from this CCG, so showing it as a proportion of the South West of England would be misleading. I have clarified throughout the text that practices come from this specific region of South West England.

	Section	Reviewer comment	Author response
			“All 82 practices in the Bristol, North Somerset and South Gloucestershire were approached by the CCG with information about the study. Practices returned expressions of interest to the researchers who selected 21 practices, aiming for a variation of deprivation, ethnic mix, practice size and locality” We have covered how representative the sample is in the discussion rather than the methods, as the reviewer suggests.
10	Data	I think the data collection and method could be explained a bit more clear. First of all, could the authors elaborate a bit about the origin of the data source? Data must have been initiated prior to the COVID since the authors can compare to 2019 data. This is such a strength and I would like to understand this a bit better when reading.	We have clarified this now. As stated, the data is routine data, from the patient record, but this, and the way it was extracted, may not have been clear. The text now says: “All participating practices then agreed for anonymised data to be extracted directly from the patient record by One Care, the GP federation in BNSSG CCG. All practices use the same EMIS electronic medical records system. One Care provide technical support all GP practices in the CCG and have the ability to extract routine data from the patient record via the EMIS enterprise search and report module. While the data was extracted retrospectively, ERP allows use of relative run dates to capture the information in the patient record at a given time in the past.”
11	Data	Second; the data selection for this paper could be explained a bit more clear. When reading it for the first time, I was confused by the different period mentioned, and I had to read it a few times to be clear when you were referring to the selection of patients (patients alive and registered with a practice 1th of July 2020) and time period included (all registered contacts from Feb 2019 until 31 July 2020). Yet,	I have added an explanation on this now: “We extracted data from February 2019 so that we had a full 18-months to examine trends for the main study. For the analyses in this paper, the period April-July 2020 (i.e. the period since the start of UK lockdown) was compared to the same period the previous year (April-July 2019).”

	Section	Reviewer comment	Author response
		why was February 2019 included when the comparison period was from April?	
12	Results	How did the authors get the 56%? From table 1, $69,254/126,466=54.8\%$ (55%). Maybe a typo? I would suggest that the authors include a total column in table 1 which would include information about the proportion of patients consulted among all registered patients per 1 July 2020. If this is included, you could skip these calculations in the result section.	The 56% is correct. 126,466 was the total number of patients in 2020. The number of patients in 2019 was 123,947. The text shows how the 56 percent was calculated $69,254/123,947$. We have added a footnote to Table 1 to clarify this.
13	Results, Page 6 line 3-7:	The authors state that the levels of consultations did not recover to prior COVID levels. Could the authors refer to a table/figure where this can clearly be assessed? (figure 2? – maybe state the proportions in the text (7% vs 4%??))	I have added this now in the text.
14	Page 6 line 31 - /table 3	I acknowledge your point with multiple testing for each symptom/ condition. But for me, I would still prefer that you e.g. marked in bold the IRR with a statistically significant reduction from 2019 to 2020. Especially among the rarer symptoms, with a very low incidence, this is relevant to assess the impact. Maybe consider using the Bonferroni correction to correct for multiple testing.	We have added the 95% confidence intervals in Table 4, and highlighted in bold those with a p-value of 0.0008 (using the Bonferroni correction to adjust for multiple testing).
15		Further, in relation to table 3 and the important issue raised by the authors in the discussion concerning the red-flag symptoms. I do acknowledge the authors choice in dividing the symptoms into “commonly/not-commonly reported symptoms”. But I wonder; in terms of cancer diagnostics, if it could bring important information to divide the symptoms according to alarm-symptoms: yes no. I would expect patients to be more likely to present an alarm symptom in a remote consultation than would be the case if the patient experienced a vague or unspecific symptom. This has great implications as the vague and unspecific symptoms often is linked with “harder to diagnose” cancer types, which also have inferior cancer prognosis in general and could be suitable to a further survival	We have now highlighted alarm symptoms in Table 4 and have made specific reference to these in the discussion. We have divided the symptoms as the reviewers suggested into alarm/non-alarm, and have added an additional row to the top of table 4 to show the difference in alarm symptoms only, as compared to all symptoms.

	Section	Reviewer comment	Author response
		disparities if these cancers are less likely to be caught during a lock down. The current categorisations of symptoms does not offer this distinction. If possible, the authors could add an analysis comparing the likelihood of presenting alarm symptom vs non-alarm symptoms during the lock-down period.	
16	Page 8	you refer to “in the over 80s” – this could be more clear throughout the section: “patients aged 80 years or older” (e.g.).	We have now changed this.
17	Table 1	Besides including a “total column” as mentioned, could you consider changing “all patients registered in July 2020” into “all patients registered per 1 July 2020”	We have adjusted this in the text but left the table as July 2020 as it may confuse the reader to have a date there.
18	Figure 2	Figure 2: Please correct the typo (‘000 should be 1000)	This has now been corrected
19		Discussion: Bias: I think the strengths and limitation section could be strengthened by referring to selection bias and information bias. The authors do to some extent cover this (You do mention that a large proportion of patients with diverse backgrounds was included, but is there reason to suspect selection bias among the selected 21 practices compared to the background practices of your study population – all practices of the UK?). You also mention the risk of information bias – without referring to is as such.	Reference to selection bias and information bias has now been added.

VERSION 2 – REVIEW

REVIEWER	Jones, Daniel University of Leeds
REVIEW RETURNED	22-Apr-2021

GENERAL COMMENTS	I think the authors have addressed all the comments made by myself and the other reviewer. I think this is a helpful paper and would be suitable for publication.
---

REVIEWER	Virgilsen, Line Aarhus Universitet
REVIEW RETURNED	27-Apr-2021

GENERAL COMMENTS	The authors have answered the raised questions satisfactory. Congratulations on the great paper.
--